# Pencils Down! Automatic Rubric-based Evaluation of Retrieve/Generate Systems

## ABSTRACT

Current IR evaluation paradigms are challenged by large language models (LLMs) and retrieval-augmented generation (RAG) methods. Furthermore, evaluation either resorts to expensive human judgments or lead to an over-reliance on LLMs.

To remedy this situation, we introduce the RUBRIC metric, which puts information retrieval systems to the proverbial test. This metric leverages a bank of query-related test questions to quantify relevant information content that is contained in the systems' responses. The process involves (1) decomposing the query into detailed questions, and (2) checking each for answerability using passages in the system response. Using three TREC benchmarks, we demonstrate that our LLM-based RUBRIC approach works successfully. Unlike previous LLM-based evaluation measures, our paradigm lends itself for incorporating a human-in-the-loop without the danger of over-reliance on AI or resorting to expensive manual passage-level judgments. Moreover, our evaluation is repeatable and extensible and can be scored with existing evaluation tools.[1]

**ACM Reference Format:**
Anonymous Author(s). 2024. Pencils Down! Automatic Rubric-based Evaluation of Retrieve/Generate Systems. In *Proceedings of ACM Conference (Conference'17)*. ACM, New York, NY, USA, 10 pages. https://doi.org/10.1145/nnnnnnn.nnnnnnn

## 1 INTRODUCTION

The advent of large language models (LLMs) has led to a plethora of information retrieval systems that combine traditional retrieval with neural ranking and natural language generation—but it is unclear how to reliably evaluate such systems. In this paper, we propose the RUBRIC evaluation paradigm which measures to which extent the systems' responses contain information content that is relevant, concise, and complete. The evaluation paradigm should take advantage of the abilities of LLMs while ensuring that human judges have the final say in determining relevance. Moreover, we develop an evaluation paradigm that is repeatable and reusable, while avoiding the need to employ human judges for tedious tasks.

In this paper we focus on the IR evaluation task with the following setup:

---

[1] **Data and code available at https://anonymous.4open.science/r/rubric/**

*Conference'17, July 2017, Washington, DC, USA*

***Task Statement: Evaluation.*** A retrieval / generation system is given a search *query q* to produce a relevant *system response*. The response can take the form of a passage ranking, a set of extractive summaries, or a single generated text—each will be considered a set of passages *P*.

Given system responses across multiple queries from multiple systems, the task is to assign each system an *evaluation score* that represents the quality of the information content provided in their responses. This evaluation score must reflect the quality with which relevant information is presented.

Traditionally, information retrieval systems are evaluated with manual assessments. This involves human judges determining the relevance of system-generated responses to specific queries. While this method is valued for its depth of insight, manually evaluating the output of retrieval systems becomes impractical as the volume of information increases. Unfortunately, restricting the scope of evaluation, will make it hard to identify subtle quality differences between systems, potentially hindering the development of more sophisticated IR approaches.

To address the drawbacks of manual evaluation, there has been a shift towards automated methods. A popular evaluation approach is to directly ask LLMs whether a passage is relevant for a query. Empirically this has been shown to work well [10, 18, 26, −*inter alia*], but skepticism remains about whether LLMs can be trusted to replicate the nuanced understanding of humans in the judgment process, especially when a deep contextual understanding of complex user needs is required. Without reliable human oversight, there is no way of knowing when this problem arises. Faggioli et al. [10] discuss many issues that arise when humans are completely removed from the evaluation process.

A significant challenge in current evaluation approaches is the lack of effective collaboration between human judges and AI [9]. Faggioli et al. [10] elaborates a wide-range of theoretical concerns, centered on questions of trustworthiness and reliability of LLMs now and in the future. Wang et al. [28] empirically demonstrate that LLMs exhibit unfair positional bias towards candidates displayed for evaluation. Liu et al. [17] demonstrate that evaluator-LLMs give a higher score to systems that are based on the same LLM. Some have suggested to ask human judges to verify an LLM's decision. However, Fok and Weld [11] have shown that human judges might over-rely on AI-generated rationales, negatively affecting their objectivity. In an opinion article, Faggioli et al. [9] suggest to better integrate humans into the definition of relevance. In this work, we develop such an approach.

*Our approach.* In light of these limitations, we propose RUBRIC– a novel approach towards evaluating information retrieval systems. Our framework integrates LLMs and human judges, establishing a division of labor that plays to the strengths of both parties. We focus on breaking down the concept of "relevance" into a grading

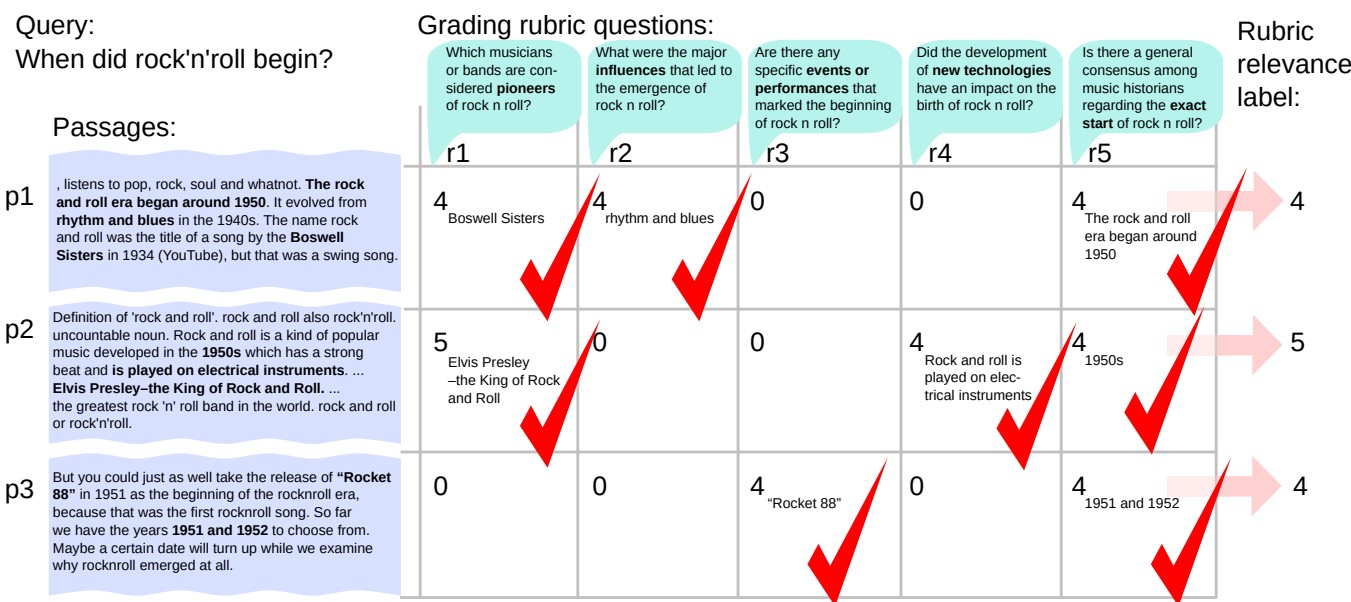

**Figure 1: The RUBRIC evaluation uses LLMs to grade how well a passage $p$ addresses each rubric question $r$. Each cell depicts the grade assigned for each passage and question on a scale from 0 (worst) to 5 (best), cf. Section 4.2, along with extracted answers for manual verification. Passage-level relevance labels for the RUBRIC evaluation score are derived from grades, to be used with traditional IR evaluation measures (Section 4.3). This example is based on actual RUBRIC grades obtained with our system for TREC DL 2020 query `940547` used in the manual verification analysis (Section 6.7).**

rubric of multiple concise questions that must be addressed in a system's response in order to be considered relevant. This yields a more structured and unbiased evaluation process. Our method is fast and efficient due to leveraging LLMs to scan all retrieved passages for answers to these test questions. At the same time, our RUBRIC paradigm puts human judges in charge of defining relevance through multiple concise test questions, thus maintaining the depth of human insight while minimizing over-reliance on AI.

*Defining relevance via grading rubrics.* We believe that the task of breaking a larger information need into the set of concise questions is more natural for humans to accomplish than to directly judge the relevance of text. The process is akin to designing a grading rubric for essay grading. Educators routinely break down complex assignments into specific criteria or questions, allowing for a more objective and detailed assessment of student work.

Similarly, in the context of IR system evaluation, decomposing each information need into distinct, answerable questions transforms the abstract concept of "relevance" into tangible criteria that can be systematically assessed. This process naturally aligns with human cognitive strengths, such as critical thinking and identifying semantic errors, enabling judges to focus on defining what constitutes relevant information through a less subjective lens.

Obviously every search query needs to be associated with its own grading rubric. But once these grading rubrics are in place, our RUBRIC framework leverages the capabilities of LLMs to conduct a systematic, replicable, and efficient evaluation of the responses retrieved and synthesized by information retrieval systems. LLMs will scan through vast amounts of retrieved material, identifying

and assessing the presence of answers to the predefined questions on the rubric. This process not only significantly reduces the time and resources required compared to manual evaluation but also ensures a consistent and objective application of the evaluation criteria across different systems and queries.

Furthermore, the use of LLMs in this capacity supports a dynamic and scalable evaluation process, that can be replicated whenever new information retrieval systems are to be tested.

To ensure that the grading rubric is complete and is indicative of relevance, our paradigm encourages human judges to inspect some system's responses along with automatic grades assigned by our evaluation paradigm. As the LLM scans system responses during grading, it can also extract free-form answers (examples in Figure 1) that may inform humans how to further refine the grading rubric, creating a feedback loop that continuously enhances the evaluation framework via a dialog between human judges and the LLM.

*Contributions.* We develop an evaluation approach that,

(1) is amenable to integrating humans and LLMs so that it plays to each of their strengths,

(2) never requires manual passage-level relevance judgments,

(3) benefits from latest advances in large language models,

(4) yields reusable test collections that can evaluate (future) systems, especially those that employ language generation,

(5) allows to expand the test collection post-hoc to reveal differences between systems.

Experimentally we demonstrate that our rubric-based evaluation approach agrees with traditional evaluation paradigms, as quantified by rank correlation of system leaderboards from three TREC test collections.

## 2 RELATED WORK

### 2.1 LLM-based Relevance Assessment

While our approach does not attempt to imitate the passage-level relevance-judgment process, several recent methods studied this approach. The idea of direct grading prompts is to ask an LLM whether a passage answers the query. We include several of these direct grading prompts as baselines in our empirical evaluation.

Sun et al. [25] uses this direct grading prompt to rerank passages. Faggioli et al. [10] produce automatic relevance labels for data from the TREC Deep Learning track. In 1SLs, MacAvaney and Soldaini [18] focus on evaluating passages with a DuoPrompt, that instructs an LLM to indicate which of two passages is more relevant for a query. Thomas et al. [26] empirically compare the ability of crowd workers and LLMs to perform document-level relevance judgments. They find that especially the label quality of crowd-workers is inferior to fully automatic LLM-based relevance labels. Thomas et al. are using a very detailed prompt (Figure 1 in [26]), clarifying the role and query description and asking the LLM to comment on the query intent and trustworthiness. We study an abridged version of this prompt in the empirical evaluation.

As discussed in the introduction, several voices raised critiques about using LLM's for relevance labels even with human supervision [10, 11, 17]. We provide an alternative to better integrate human judges into this process.

### 2.2 Nugget-based Evaluation

There is a long history in the IR community to evaluate the relevance of documents by breaking down the information need into a set of "nuggets" (also called query intents, facts, or SCUs) that can each be evaluated independently [16]. The common definition of a nugget is "the smallest portion of text that constitutes relevant information in and of itself" [21].

With the advent of LLMs, nugget-style evaluation is being revamped, most recently in the TREC Crisis Facts track [19]: Judges are asked identify atomic "facts" (similar to nuggets). System responses are analysed for mentions of these facts, either via a boolean OR or with an embedding-based method.

Our proposal is related in that we break down the information need into a set of rubric elements that represent relevance. In Section 5.1 we discuss a variation of our approach that use rubrics of nuggets instead of questions.

### 2.3 Evaluation with Test Questions

The idea of basing an evaluation on a bank of test questions has been widely discussed in literature on summarization [3]. Eyal et al. [8] suggest an system evaluation score that is based on the number of questions that a question answering system can correctly answer using the system response—a principle that our approach follows.

Many approaches use a Cloze-style idea to generate questions from a given gold summary or source text, generating multiple-choice questions [12], questions with exact-match answer verification [6], or entities-centric questions [8, 27].

In information retrieval evaluation there is no source text or gold summary to generate questions from. Sander and Dietz [23] avoid this problem in the EXAM Answerability Metric by using human-designed multiple-choice test questions that would indicate relevance for the search queries of TREC CAR Y3. They use a question answering system to automatically check whether system responses can answer their test questions.

Where EXAM uses a pre-neural question answering system that is limited to multiple-choice questions, our RUBRIC approach builds on the advent of modern LLMs to permit open-ended questions. Additionally, we offer automated support for creating test questions.

### 2.4 LLMs, Passages, and Questions

Many approaches integrate passages, questions, and LLMs in some form. This includes improving question answering via retrieval-augmentation [14, 22]. Improve fact verification, by breaking each claim down into several questions [30]. Exploiting the self-verification ability of LLMs to improve the reasoning [29]. Evaluating the quality of LLMs with multiple tests [2, 15]. Arabzadeh et al. [1] develop an approach to improve LLM embeddings, by generating "liar" questions that cannot be answered with the given context.

## 3 APPROACH OVERVIEW

Our approach is based on the idea of developing a grading rubric for each query: a set of questions that a good system response should be able to answer. Our proposed RUBRIC metric quantifies the coverage and quality of relevant information content provided in system responses. These responses could be retrieved from a corpus, generated from scratch, or generated with retrieval-augmentation. The systems are graded based on query-specific grading rubrics of test questions, tracking which rubric elements are addressed and how relevant, complete, and accurate the provided answer is. The more test questions can be addressed well, the higher the RUBRIC evaluation score of the system.

We remark that the grading rubrics are intended to be secret: systems under evaluation should not have access to the grading rubric when responding to the search query.

*Inputs.* Our RUBRIC evaluation system assumes the following inputs:

(1) A set of queries, optionally with query subtopics.
(2) A set of system responses, which can come in the form of a passage ranking or a list of generated passages.
(3) If available, a grading rubric for each query to be refined.

*Outputs.* As part of the evaluation, our approach will operate in three phases (depicted in Figure 1), creating the following outputs:

**1. Designing grading rubrics:** A process of creating a rubric of test questions for each query, each question representing one important piece of information that should be addressed in the system's response. Our framework supports grading rubrics that expect unstructured answers as well as those verifiable

**Table 1: Question generation prompts (GenR). The prompt includes instructions to respond in JSON format for easier parsing.**

| Question Generation: TREC DL Prompt | Question Generation: TREC CAR Y3 Prompt |
| --- | --- |
| Break the query '{query_title}' into concise questions that must be answered. Generate 10 concise insightful questions that reveal whether information relevant for '{query_title}' was provided, showcasing a deep understanding of the subject matter. Avoid basic or introductory-level inquiries. Keep the questions short. | Explore the connection between '{query_title}' with a specific focus on the subtopic '{query_subtopic}'. Generate insightful questions that delve into advanced aspects of '{query_subtopic}', showcasing a deep understanding of the subject matter. Avoid basic or introductory-level inquiries. |

**Table 2: Grading prompts.**

| Grading: Self-rating Prompt |
| --- |
| Can the question be answered based on the available context? choose one: 
 - 5: The answer is highly relevant, complete, and accurate. 
 - 4: The answer is mostly relevant and complete but may have minor gaps or inaccuracies. 
 - 3: The answer is partially relevant and complete, with noticeable gaps or inaccuracies. 
 - 2: The answer has limited relevance and completeness, with significant gaps or inaccuracies. 
 - 1: The answer is minimally relevant or complete, with substantial shortcomings. 
 - 0: The answer is not relevant or complete at all. 
 Question: {question} Context: {context} |

with gold standard answer keys. While human judges should design the grading rubric, if desired, the rubric creation can be seeded with automatically generated grading rubrics. This is discussed in Sections 4.1.

2. **Graded system responses:** All passages in system responses are automatically graded via an LLM: Each passage is scanned for information content that addresses each rubric element, assessing the quality of the provided information on a scale from 0 (worst) to 5 (best), as elaborated in Section 4.2.

3. **RUBRIC evaluation scores:** Our approach scores systems with an evaluation score that is based on how well rubric elements are addressed in the system's response. Our RUBRIC metric derives a relevance label for each passage, based on the best addressed rubric element and computes each system's evaluation score with `trec_eval` based on these relevance labels as described in Section 4.3.

*Human-in-the-loop.* We envision that human judges focus their efforts on designing grading rubrics (Phase 1), while LLMs are automatically grading system responses (Phase 2). Next, human judges should inspect some the grading results to improve the grading rubric by reformulating, adding, or deleting questions and adjusting the LLM's prompt/few-shot exemplars/fine-tuning setup to provide more accurate grades. Once this process is complete, system evaluation scores are computed based on the graded responses, in a fashion similar to `trec_eval`.

While our workbench software is designed to incorporating a human-in-the-loop,[2] in this article we focus on the feasibility of the automatic part of this evaluation paradigm and demonstrating the verification process in Section 6.7. We leave the human subject study to future work.

*Reusability of test collections.* As new systems are developed, these can be graded and evaluated with the developed grading rubrics. By dividing the evaluation process into rubric generation and automatic grading, our approach avoids the problem of unjudged passages (also called "holes" [18]) in test collections. RUBRIC grading pipeline can be applied to update the "qrels" file whenever new passages are retrieved or generated by new systems.

This process allows our evaluation paradigm to be easily incorporated into shared tasks of evaluation venues like TREC, NTCIR, or CLEF, as the only the "qrels" file needs to be distributed to participants to develop systems with the RUBRIC metric.

*Extensibility of test collections.* Traditionally, test collections are created by pooling system responses, and then frozen once completed. However, as increasingly better systems are developed, these may obtain new information content that should have been incorporated in the grading rubric, but were previously not known by human judges. The RUBRIC evaluation paradigm readily supports modifying the grading rubric in light of new information, to be deployed as an updated version of the test collection.

Below in Section 4, we describe the best performing implementation of this RUBRIC framework, before detailing alternative implementations and baselines in Section 5 which are included in the experimental evaluation.

## 4 GENERATED RUBRICS FOR SELF-RATINGS

### 4.1 Phase 1: Generated Grading Rubrics

While human judges should focus on the rubric creation task, our system can provide an initial seed rubric for each query via a generative LLM.

In our experiment, we use GPT 3.5 to obtain initial grading rubrics (to be refined by human judges). The prompt is designed to elicit a set of concise, insightful test questions $r$ based on the query, tailored to specific goals of the IR task and domain. For TREC DL we ask to break the question query into concise sub-questions. In application to TREC CAR Y3, we ask for questions that explore the connection between the broad query with a specific focus on each

---

[2]URL to online appendix in abstract.

**Query title:** When did rock'n'roll begin?

$r_1$ Which musicians or bands are considered pioneers of rock n roll?

$r_2$ What were the major influences that led to the emergence of rock n roll?

$r_3$ Are there any specific events or performances that marked the beginning of rock n roll?

$r_4$ Did the development of new technologies have an impact on the birth of rock n roll?

$r_5$ Is there a general consensus among music historians regarding the exact start of rock n roll?

$r_6$ Did rock n roll start as a distinct genre or did it evolve from existing music styles?

$r_7$ Were there any notable recordings or songs that played a significant role in popularizing rock n roll?

$r_8$ What cultural and social factors contributed to the rise of rock n roll?

$r_9$ Did rock n roll have regional variations or was its impact worldwide?

$r_{10}$ Were there any significant changes in the music industry that paved the way for rock n roll?

**Figure 2: Generated grading rubric for TREC DL 2020 query 940547, of which Figure 1 displays r1–r5.**

**Query title:** The Integumentary System
**Query subtopic:** Structure of the Skin

r1 How does the epidermis, dermis, and hypodermis work together to provide protection, sensation, and regulation for the body?

r2 Can the integumentary system be compromised by diseases and conditions, and if so, how does this impact the health of the skin?

r3 How does the skin act as a barrier against pathogens and other foreign substances?

**Passage**: b95bf325b7fdacac183b1daf7c118be407f52a3a
The skin is the largest organ in the human body. Skin is made up of three layers, the epidermis, dermis and the fat layer, also called the hypodermis. ==The epidermis is the outer layer of skin that keeps== vital fluids in and ==harmful bacteria out of the body. The dermis is the inner layer of skin== that contains ==blood vessels==, ==nerves==, hair follicles, oil, and ==sweat glands==. Severe damage to large areas of skin exposes the human organism to dehydration and infections that can result in death.
**TREC judgment**: 3 (MUST be mentioned)

**Figure 3: Excerpt of generated grading rubric for TREC CAR Y3 query `tqa2:L_0384`. Matching text spans highlighted in passage.**

subtopic. The complete prompts used in the experimental evaluation are listed in Table 1.

From these prompts we obtain test questions asking for unstructured free-form answers, such as given in Figure 2.

## 4.2 Phase 2: RUBRIC Grading with Self-Ratings

In this phase, we use an LLM to identify all relevant material in passages of system's responses. We consider each each question $r$ on the rubric and track results per passage $p$ as $\text{grade}(r, p)$, a numerical grade.

To initialize the grading phase, we pre-process system responses to obtain a set of paragraph-sized plain text passages, each up to 400 tokens in length (associated with a unique `passage_id`).

*4.2.1 Grading by self-rating.* Each $\text{grade}(r, p)$ quantifies how well rubric question $r \in R_q$ is addressed in passage $p \in P$ on a scale from 0 (worst) to 5 (best), as depicted in Figure 1. We lean on the ability of modern LLMs to match language patterns and ask the LLM to self-rate the answerability of question $r$ using each passage $p$ as context using the grading prompt given in Table 2.

Critical elements of the prompt is to ask "how relevant, complete, and accurate" the answer is, and to emphasize that we seek answers with the available context. The exact phrasing of the prompt was suggested by GPT 4 for use with the `FLAN-T5-large` model.

We observe that LLMs are very reliable when matching concise information content according to the grading rubric, a phenomenon Weng et al. [29] calls self-verification behavior. We find that in the vast majority of cases, modern LLMs, such as `FLAN-T5-large`, indeed respond with a numerical code between 0 and 5. In the remaining (rare) cases, we assign a rating of 1 by default or 0 when expressions of unanswerability are encountered.[3]

To support human judges to oversee this process, we complement the numerical self-rating grade with an extracted textual answer (depicted in small font in Figure 1). In the manual verification (cf. Section 6.7), we find that numerical grades mostly line up with extracted answers.

In contrast to Sander et al. [23], who evaluate answerability with multiple-choice questions, our process avoids many technical difficulties in matching gold answers in the light of different ways to phrase a correct answer. As demonstrated in Figure 1, many different answers are correct and extracting such answers is helpful for manual verification. Furthermore, if desired, the extracted answer can also be used to complement the self-rating process with verification of the LLMs's answer against the gold answer key.

## 4.3 Phase 3: RUBRIC-based Evaluation Metrics

Based on the grades of each passage/question combination, we can derive a RUBRIC evaluation score for each system, which is averaged across all queries in the test set.

For each query, we associate each passage $p$ with a relevance label according to the best grade achieved on any of the rubric elements $r$.

$$\text{relevance-label}(p) = \max_{r \in R_q} \text{grade}(r, p) \qquad (1)$$

We expose these RUBRIC-based relevance labels as a `trec_eval` compatible relevance file (aka "qrels" file). This permits to implement our novel Rubric Score evaluation metric with an established evaluation tool-chain such as `trec_eval`, building on traditional evaluation metrics. By configuring `trec_eval` to use the multi-relevance grading threshold[4] $\tau$, we only count passages as relevant that obtain at least a minimum grade of $\tau$ on any of the rubric elements. Moreover, our provided software can be configured to emit a relevance label based on the best grade achieved by at least $m$ rubric elements instead of just the best.

Empirically we find that the RUBRIC evaluation metric obtains a high correlation with official leaderboards of all three test sets.

---

[3]"unanswerable", "no", "no answer", "not enough information", "unknown", "it is not possible to tell", "it does not say", or "no relevant information".
[4]set with `trec_eval` option `--level_for_rel`

## 5 RUBRIC VARIATIONS AND BASELINES

In this section we elaborate an alternative variation as well as some state-of-the-art baselines we explore in the empirical evaluation in Section 6.

### 5.1 Variation: Nugget-based Grading Rubrics

Instead of basing the grading rubric on questions, we can build on work of nugget-based evaluation [16, 21, 24] and create grading rubrics $R_q$ comprised of nuggets or key facts.

The only difference lies in the changing prompts. For generating nugget-based grading rubrics, the prompts listed in Table 1 need to be adjusted to ask for "key facts" instead of questions. For grading in Phase 2, the prompts in Table 2 need to ask "to which extent a key fact is covered" in the given passage. (We list the complete prompts in the online appendix.)

In the experimental evaluation (Section 6.2), we find that the nugget-based approach works less well. We believe the main reason is that LLMs are trained on a wide range of question answering benchmarks, but only very few test collections with nuggets.

### 5.2 Baseline: EXAM Metric

The EXAM method [23] uses a pre-neural question answering system from AI2, which was designed to answer multiple-choice questions with given context for questions in the style of the ARC and TQA datasets. The answer is verified as correct when the question answering system responds with the correct choice, resulting in binary grades per passage and question. The downside is that this question answering system was difficult to set up,[5] In contrast, our RUBRIC approach uses a modern LLM-based question answering system that is easy to integrate.

### 5.3 Baseline: Direct LLM-Grading

A very competitive approach is to ask an LLM whether a passage $p$ is relevant for a given query $q$—without intermediaries such as test questions or nuggets [10, 20, 25, 26]. These methods have been empirically shown to be very capable, hence we include these in the experimental evaluation as an upper-bound reference. However, as described above [11, 17], it is not possible to incorporate the humans into this direct grading paradigm without (1) the danger of judges' over-reliance on AI during verification [11] or (2) the need for manual passage-level judgments.

## 6 EXPERIMENTAL EVALUATION (OF THE EVALUATION METRIC)

### 6.1 Experimental Setup

*Evaluation methodology.* We study our approach on three TREC datasets by providing an alternative evaluation of systems submitted to the respective TREC tracks. We demonstrate that our method reproduces the official leaderboard results. A higher rank correlation in terms of Kendall's tau and Spearman's rank correlation coefficient implies better evaluation paradigm.

Additionally, we compare our automatically predicted passage relevance labels to manually produced official TREC judgments,

in terms of count statistics and of Cohen's kappa inter-annotator agreement which corrects for chance agreement.

*Datasets.* We use the following test collections:

**TREC DL 2019 [5]:** Using 43 queries in the question-form from the Deep Learning track, harvested from search logs. The system's task is to retrieve passages from a web collection that answer the query. The official track received 35 systems, metrics are NDCG@10, MAP, and MRR.

**TREC DL 2020 [4]:** Similar setup as the previous Deep Learning track, but with 54 additional queries and 59 submitted systems.

**TREC CAR Y3 [7]:** Comprising 131 queries and 721 query subtopics from the TREC Complex Answer Retrieval track. These were harvested from titles and section headings from school text books provided in the TQA dataset [13]. The system's task is to retrieve Wikipedia passages to synthesize a per-query response that covers all query subtopics. Official track metrics are MAP, NDCG@20, and R-precision; of 22 systems were submitted to this track, several have identical rankings. We use 16 distinguishable systems used by Sander et al.

### 6.2 Compared Evaluation Methods

We compare several variations of our RUBRIC paradigm as well as a range of established baselines.

**RUBRIC:** Represents our proposed implementation described in Section 4 using generated grading rubrics (Phase 1), grading with self-ratings (Phase 2), to derive RUBRIC relevance labels. We obtain ten questions for each query-subtopic in TREC CAR Y3, and each query in TREC DL.

**Nugget RUBRIC:** Same as RUBRIC, but using a rubric of nuggets instead of questions.

While any LLM can be used for grading in our evaluation paradigm, in this work we focus on affordable LLMs and use GPT 3.5[6] for rubric generation and the recent `FLAN-T5-large` model with the text2text-generation pipeline from Hugging Face.[7] This allows to complete RUBRIC grade annotations for TREC DL 2019 within 1 hour on an NVIDIA A40 GPU.

We grade all passages in official judgments and the top 20 of all submitted system runs. This results in 85,329 passages in TREC CAR Y3, 9,260 passages in TREC DL 2019, and 11,386 passages for TREC DL 2020.

**Baselines.** To demonstrate the quality of our approach we compare to several established baselines.

**EXAM [23]:** Using a pre-neural question answering system on the multiple choice question from the TQA dataset, as described in Section 4.3. Results are taken from the original the paper (available for TREC CAR Y3 only).

Additionally we compare to the following direct grading prompts (cf. Section 5.3) using the same LLM as above (`FLAN-T5-large`).

**Thomas [26]:** "Instruction: You are a search quality rater evaluating the relevance of passages. Given a query and a passages, you must provide a score on an integer scale of 0 to 2 with the

---

[5]We were unable to install the question answering system used in the original EXAM method.

[6]`gpt-3.5-turbo-instruct`
[7]https://huggingface.co/google/flan-t5-large

**Table 3: Rank correlation of Rubric Score with the official leaderboard compared to RUBRIC, nugget variation, direct LLM grading prompts, and original EXAM method. S: Spearman's rank correlation. K: Kendall's Tau correlation—since both measures display the same characteristics, some are omitted. Evaluation measures chosen to match dataset recommendations [4, 5, 7]. More results in online appendix. Best results per column denoted in bold-italic, equally good methods denoted in bold. "-" denotes results that cannot be computed as no grades above $\tau$ were obtained. Note: `trec_eval`'s NDCG reports identical results for different settings of `--level_for_rel`.**
**Our proposed RUBRIC approach reliably obtains best results, which are as good or slightly better than direct grading approaches, while offering a clear path for integrating human oversight into the process.**

| | | TREC DL 2020 | | | | TREC DL 2019 | | | | TREC CAR Y3 | | | | Wins |
| | | NDCG@10 | | MAP | MRR | NDCG@10 | | MAP | MRR | MAP | | NDCG@20 | RPrec | |
| Evaluation | $\tau$ | S | K | S | S | S | K | S | S | S | K | S | S | |
| RUBRIC | 1 | *0.974* | **0.875** | 0.846 | 0.865 | **0.961** | **0.848** | 0.440 | **0.850** | **0.931** | 0.808 | *0.883* | **0.909** | 8 |
| | 3 | *0.974* | **0.875** | 0.872 | **0.937** | **0.961** | **0.848** | 0.418 | 0.731 | **0.933** | 0.817 | *0.883* | 0.910 | 8 |
| | 4 | *0.974* | **0.875** | 0.893 | *0.941* | **0.961** | **0.848** | 0.467 | 0.696 | **0.933** | 0.817 | *0.883* | 0.910 | 8 |
| Best overall→ | 5 | *0.974* | **0.875** | *0.946* | 0.845 | **0.961** | **0.848** | *0.882* | 0.795 | *0.980* | *0.902* | *0.883* | *0.959* | 10 |
| Nugget RUBRIC | 1 | **0.947** | 0.802 | 0.626 | 0.524 | **0.969** | **0.856** | -0.152 | 0.423 | 0.920 | 0.789 | **0.848** | **0.915** | 5 |
| | 3 | **0.947** | 0.802 | 0.681 | 0.275 | **0.969** | **0.856** | 0.057 | 0.526 | 0.873 | 0.721 | **0.848** | 0.907 | 4 |
| | 4 | **0.947** | 0.802 | 0.817 | 0.609 | **0.969** | **0.856** | 0.355 | 0.598 | 0.876 | 0.762 | **0.848** | 0.893 | 4 |
| | 5 | **0.947** | 0.802 | **0.940** | 0.838 | **0.969** | **0.856** | 0.858 | 0.798 | 0.894 | 0.747 | **0.848** | 0.878 | 6 |
| Thomas [26] | 1 | **0.936** | 0.810 | 0.828 | 0.751 | **0.960** | 0.833 | 0.341 | 0.755 | 0.666 | 0.576 | 0.640 | 0.646 | 2 |
| FagB [10] | 1 | **0.966** | **0.861** | 0.922 | **0.940** | 0.968 | **0.875** | **0.864** | **0.810** | 0.588 | 0.443 | 0.582 | 0.685 | 8 |
| FagB_few [10] | 1 | **0.970** | **0.872** | 0.924 | 0.918 | *0.979* | **0.885** | 0.859 | 0.771 | 0.284 | 0.179 | 0.409 | 0.320 | 7 |
| HELM [15] | 1 | **0.970** | **0.872** | 0.919 | 0.930 | 0.962 | 0.851 | 0.863 | 0.829 | 0.550 | 0.434 | 0.486 | 0.520 | 8 |
| Sun [25] | 1 | *0.974* | *0.880* | 0.920 | 0.924 | 0.948 | 0.828 | 0.823 | 0.757 | 0.655 | 0.510 | 0.627 | 0.677 | 5 |
| Sun_few [25] | 1 | **0.950** | 0.825 | **0.928** | 0.866 | *0.979* | *0.894* | *0.882* | *0.852* | 0.286 | 0.180 | 0.286 | 0.175 | 6 |
| EXAM [23] | | | | | | | | | | 0.75 | 0.57 | 0.74 | 0.74 | |

following meanings:

2 = highly relevant, very helpful for this query

1 = relevant, may be partly helpful but might contain other irrelevant content

0 = not relevant, should never be shown for this query
Question: {query_title} Passage: {context} Answer:"

**FagB [10]:** "Instruction: Indicate if the passage is relevant for the question. Respond with 'Yes' or 'No'.
Question: {query_title} Passage: {context} Answer:"

**FagB_few [10]:** FagB with additional few shot prompting.

**HELM [15]:** "Instruction: Does the passage answer the query? Respond with 'Yes' or 'No'.
Question: {query_title} Passage: {context} Answer:"

**Sun [25]:** "Instruction: Given a passage and a query, predict whether the passage includes an answer to the query by producing either "Yes" or "No".
Question: {query_title} Passage: {context} Answer:"

**Sun_few [25]:** Sun with additional few shot prompting.

Since Sander's work demonstrated that ROUGE metrics are uncorrelated with leaderboard rankings, we omit the comparison here.

**Leaderboard Correlation.** We compare different evaluation paradigms by how well their leaderboards correlate with the official leaderboard of respective datasets. Relevance labels are derived from RUBRIC grades and the direct grading prompts, to be used as "qrels" files. Each system is scored using `trec_eval` with these relevance labels using official evaluation metrics recommended for each dataset. The leaderboard ranks all systems based on their evaluation score. The correlations between such leaderboards and the official leaderboard of the TREC track is measured with Spearman's rank correlation and Kendall's tau correlation. Both correlation measures range from -1 (worst) to 1 (best) with 0 referring to uncorrelated leaderboards. We use a `scikit-learn` implementation of both rank correlation measures, where tied systems are assigned the average of their ranks.

**Significance testing.** Significance tests do not apply to these rank correlation measures. Instead we measure standard error bars for each system's evaluation score, then estimate their impact on the rank correlation results, which is about ± 0.05. Best methods marked in bold-italics, methods within that range are considered equally good (marked in bold).

## 6.3 Leaderboard Correlation on TREC DL

Table 3 presents how well system rankings on the leaderboard under each evaluation metric correlated with the official leaderboard of TREC DL. We evaluate the generated relevance labels using official track metrics normalized discounted cummulative gain (NDCG@10),[8] average precision (MAP), and reciprocal rank (MRR). Since both Spearman's and Kendall's rank correlation results paint the same picture, we omit some cases here, but provide full results in the online appendix.

---

[8]While for NDCG, `trec_eval` uses multi-relevance grades, the grading threshold $\tau$ is ignored, yielding same evaluation score.

Across all evaluation results in both 2019 and 2020 test sets, we find that our proposed RUBRIC method is consistently among the best performing metrics (e.g., 6/8 wins for self-ratings of $\tau = 5$). For illustration, we provide an excerpt of the RUBRIC leaderboard for TREC DL 2020 in Table 4.

We find that using question-based rubrics obtains slightly better results than nugget-based rubrics (cf. Nugget RUBRIC). We suspect that this is due to the abundance of question answering datasets used to train LLMs, while only few datasets with nuggets are available.

Many methods obtain Spearman rank correlations above 0.9 (a metric ranging from -1 to +1), indicating that empirically all these LLM methods are strong contenders for IR evaluation paradigms. However, it is more natural to integrate a human judge in the RUBRIC paradigm.

## 6.4 Evaluating Text Generation Systems

We demonstrate that our RUBRIC approach can be used to evaluate systems that use natural language generation. We use GPT-4 and 3.5 to develop six systems that generate system responses in response to all TREC DL 2020 queries using the following prompts.

**GPT\*-wiki:** "Generate a 1000-word long Wikipedia article on {query_title}"

**GPT\*-web:** "Generate a web page for {query_title}"

**GPT\*-question:** "{query_title}?"

These systems were not submitted as systems to the TREC track, and hence, were not manually assessed. The evaluation results are integrated in Table 4, marked with "*". We demonstrate that MRR-based Rubric Score is able to compare these methods on methods on the official leaderboard. The ranks are shifted, because the official leaderboard does not rank our six methods.

We observe that our GPT-question methods are placed on top of the leaderboard, while the GPT-web methods are placed below rank 52, GPT-wiki place around rank 40. When using a recall-based measure such as MAP, we find that GPT-wiki based method place best, as these responses cover a wide-range of facts (results omitted). We conclude that despite our GPT methods not participating in the TREC judgment pool, we can observe their relative value. Furthermore, we find that only few submitted systems swap ranks between these two leaderboards.

## 6.5 Inter-Judge Agreement on TREC DL

We analyze the grade/judgment agreement between manual TREC DL 2020 judgments and predicted relevance labels in Table 5. Cohen's $\kappa$ inter-annotator agreement referring to boxed cells, confirming a good per-passage correlation. According to track guidelines [5], judgment level 1 indicates a non-relevant passage.

Tallying each relevance label against judgments, we demonstrate that relevant judgments (2 and 3), correlate the highest with a grade of 4, and judgment level 0 correlate the highest with a grade of 0. We also observe this in the manual verification (Section 6.7).

When collapsing RUBRIC grades to a binary scale (4 and 5 relating to relevant judgments 2 and 3) we confirm good correlation with a Cohen's $\kappa$ of 0.25. We compare to the direct grading prompt of Sun et al. [25], which is obtains best NDCG@10 on TREC DL 2020 (Table 3). Their prompt yield slightly lower Cohen's $\kappa$ of 0.23

**Table 4: Leaderboard for TREC DL 2020 using RUBRIC with self-rating threshold $\tau = 4$ and MRR versus official ranks. Additional GPT-based text generation methods are included by us (denoted with \*).**

| Method | GPT | RUBRIC MRR | RUBRIC rank | Official rank |
|---|---|---|---|---|
| *GPT4-question* | * | *0.75* | *1* | |
| *GPT3.5-question* | * | *0.74* | *2* | |
| pash_f3 | | 0.74 | 3 | 3 |
| | | | … | … |
| bigIR-T5xp-T5-F | | 0.63 | 38 | 27 |
| *GPT3.5-wiki* | * | *0.63* | *40* | |
| TUW-TK-2Layer | | 0.62 | 41 | 34 |
| | | | … | … |
| terrier-InL2 | | 0.54 | 44 | 44 |
| *GPT4-wiki* | * | *0.53* | *46* | |
| terrier-BM25 | | 0.53 | 47 | 45 |
| | | | … | … |
| TF_IDF_d_2_t_50 | | 0.51 | 51 | 53 |
| *GPT3.5-web* | * | *0.51* | *52* | |
| p_bm25rm3 | | 0.50 | 53 | 49 |
| | | | … | … |
| indri-lmds | | 0.48 | 57 | 47 |
| *GPT4-web* | * | *0.47* | *58* | |
| terrier-DPH | | 0.45 | 59 | 52 |
| | | | … | … |
| DoRA_Large | | 0.11 | 67 | 59 |

according to our reproduction. In comparison between these methods our method misses 272 relevant passages, but avoids to assign a relevant label to 1115 non-relevant passages.

Hence, we confirm that RUBRIC is a competitive method. We expect to see further improvements when humans are integrating into this paradigm.

## 6.6 Results on TREC CAR Y3

As displayed in Table 3, our proposed RUBRIC method continues to provide strong results on the TREC CAR Y3 dataset, obtaining near-perfect correlation with the official leaderboard and a good grade/judge inter-annotator agreement of 0.38 (Table 6).

In contrast, LLM-based direct grading prompts Sun, FagB, HELM, and Thomas, which were head-to-head on TREC DL, are now dropping to a Spearman's rank correlation below 0.7. We speculate that the broad topical queries of the CAR collection, benefit from breaking the information need into different questions that can be verified individually.

Furthermore, we outperform Sander's EXAM method [23] by lieu of a modern LLM-based grading method that can handle open-ended questions. In their paper, Sander remarks that the rank correlation between MAP and RPrec metrics based on the official judgments obtains about 0.94 of Spearmans' and 0.86 in Kendall's tau rank correlation. We point out relevance labels produced that our RUBRIC approach reach this level as well.

**Table 5: Grade/judgment inter-annotator agreement on TREC DL 2020. Comparing RUBRIC to best method in terms of NDCG@10 (Sun [25]). Cohen's $\kappa$ referring to the boxed cell. Highest count per column is marked in bold.**

| | Grade | Judgments | | | | Total | Cohen's $\kappa$ |
|---|---|---|---|---|---|---|---|
| | | 3 | 2 | 1 | 0 | | |
| RUBRIC | 5 | 64 | 87 | 80 | 276 | 507 | |
| | 4 | **325** | **522** | 720 | 1301 | 2868 | 0.1 |
| | 3 | 23 | 35 | 61 | 255 | 374 | |
| | 2 | 14 | 54 | 120 | 299 | 487 | |
| | 1 | 4 | 14 | 17 | 75 | 110 | |
| | 0 | 216 | 308 | **942** | **5574** | 7040 | 0.29 |

| | Grade | Judgments | | Total | Cohen's $\kappa$ |
|---|---|---|---|---|---|
| | | 2–3 | 0–1 | | |
| RUBRIC | 4–5 | **998** | 2377 | 3375 | 0.25 |
| | 0–3 | 668 | **7343** | 8011 | 0.25 |
| Sun | 1 | **1272** | 3492 | 4764 | 0.23 |
| | 0 | 394 | **6228** | 6622 | 0.23 |

**Table 6: Grade/judgment agreement on TREC CAR Y3.**

| | Grade | Judgments | | Total | Cohen's $\kappa$ |
|---|---|---|---|---|---|
| | | 1–3 | -2–0 | | |
| RUBRIC | 4–5 | **1910** | 1117 | 3027 | 0.38 |
| | 0–3 | 880 | **2445** | 3325 | 0.37 |

Figure 3 depicts how three of the generated questions are addressed by a passage that was manually judged as highly relevant by TREC assessors.

### 6.7 Human-in-the-Loop Verification

For TREC DL 2020 query 940547, we analyzed 20 high ranked passages with all 10 test questions and manually verified the resulting 200 automatically assigned grades. The average rubric grade is 2.48, which coincides with the average grade on the range from 0 (worst) to 5 (best). We find that most of the time either grade 0 or 4 is awarded—grade 5 only 24 times.

The grade distribution differs per test question, with passages being awarded generally a higher grade on the test questions of widely mentioned facts such as music styles (3.9) and pioneers (2.7). As expected, some rubric questions that address less prevalent facts obtain a lower grade across all passages. This is the case for the rubric question on social factors (average grade 1.8) and impact of technological developments (0.8).

Focusing on relevant rubric grades (4 and 5), in about 75 cases the extracted answer was indeed a correct interpretation of the passage and the rubric question (versus 41 incorrect). Questions about pioneers, influences, events, and recordings are nearly perfectly answered, whenever the answer was contained in the passage.

Most of the extraction mistakes are due to misinterpretations of the question, for example for the question about whether rock'n'roll

evolved from existing music styles, for 8 of 20 passages the extracted answer said "rock n roll"—a non-answer. For the question on whether its impacts were worldwide, for 16 passages the extracted answer is "worldwide" without the passage elaborating this fact. We suspect in the latter case, the LLM is answering this question from memory instead of the provided context. In the case of several mistakes we found that the passage indeed discussed the question (deserving a high grade), but the extracted answer was incorrect.

To identify spurious test questions we count how often a negatively judged passages obtains a positive grade for a question. In this example, spurious questions are whether rock'n'roll evolved from existing styles (116 passages) and about the exact start of rock'n'roll (102 passages).

Furthermore, we can analyze passages with relevant judgments that are not associated with a positive grade. This would imply that additional questions should be added to the grading rubric. In the example above, no such passages exist.

However for query 1108651 "what the best way to get clothes white", we find a few relevant passages about bleach, that technically did not answer the rubric question "How does soaking clothes in bleach affect their whiteness?". This question would be better reformulated to "Will bleach turn clothes white?"

Detailed data of this verification is available in the online appendix (URL in abstract).

## 7 CONCLUSION

With RUBRIC we are proposing an alternative evaluation approach that integrates human judges, but not to create or verify manual passage-level relevance judgments. Instead, a grading rubric is created as part of the topic development, envisioning that each question addresses one important piece of information content. As a result, whenever such questions are answerable with responses from a retrieval / generation system, we conclude that the system provides relevant information.

Using three TREC data sets, we demonstrate that (1) our proposed approach can reproduce official TREC leaderboards nearly perfectly (Spearman's rank of 0.97); and (2) it is a strong contender in comparison to other recent LLM-based relevance label predictors [10, 15, 25, 26]. In contrast, RUBRIC offers a clear path towards integrating a human-in-the-loop, by supporting the refinement of the grading rubrics, which is how relevance is defined by judges.

We believe that further research will improve the question bank generation, as well as study the positive effects of this approach on the quality, cost, and satisfaction of human judges. In this work we demonstrate that this approach is a worthwhile option to consider.

We hope that by providing an easy means to use the RUBRIC evaluation metric via trec_eval, we offer an evaluation system that can be easily adopted by future IR evaluation tracks, offering organizers an avenue to reduce assessment costs and to obtain reusable test collections for generative information systems.

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
