# OpenReview forum: "Pencils Down! Automatic Rubric-based Evaluation of Retrieve/Generate Systems"
_ACM.org/SIGIR/ICTIR/2024/Conference — ICTIR 2024_

### Official Review · Reviewer_ej3k · 2024-05-12

**Rating:** 1
**Confidence:** 4

**Objective Part Of Review:**

Strengths:
- The problem is very important and timely.
- The motivation of the paper is well described.
- The related work section is relatively complete.
- The GitHub is well documented.

Comments:
- The authors assert that their evaluation methodology does not require passage-level relevance judgments. However, this could potentially overlook errors made by the LLM when labeling passages for different subtopics. This seems to contradict the initial claim that the methodology allows interaction between humans and LLMs. While automation eliminates the need for manual labeling, it might also reduce our confidence in these labels.

- it would be beneficial to review the distribution of graded passages in Table 2 e.g., if the passages were labeled in a balanced manner or if certain levels of relevance were scored disproportionately. This could guide adjustments in the granularity of the framing i.e., do we need more fine-grained levels? or can have less number of different grades?

- The method of integrating the relevance label of a passage through the maximum function in Equation 1 is questionable. For example, if one passage scores a level 4 across all rubrics and another scores one "5" and the rest zero, it's debatable whether the latter is more relevant.

- The robustness and redundancy of the grading rubrics are not thoroughly explored. While it might be challenging to run extensive tests on this, repeating the experiments could provide valuable insights. Excessive grading rubrics could increase costs without significantly enhancing information value.

- Most importantly, it's critical to understand what new insights the evaluation strategy provides beyond merely matching the order of the leaderboard. Why should this methodology be preferred over traditional IR evaluation metrics if it only replicates existing leaderboards?

**Subjective Part Of Review:**

- It is unclear why the authors limited the number of subquestions to 10, as not all queries may require the same number of subquestions.

- Why did the authors decide to to conduct experiments in a zero-shot setting. The results in Table 1 and for the prompts could potentially differ with in-context learning examples.

- The paper heavily references the Trec_eval tool in section 3.3, which might limit its applicability. There are other IR evaluation tools that would be used for implementation. I think these details should have moved to implementation details and the approach overview should not be limited to a tool.

- The authors mention the "best performing implementation" of the RUBRIC framework in section 3 without clear justification for this designation.

- The use of different LLMs throughout the experiments (gpt to flan-T5 )  is inconsistent, raising methodological concerns.

---

### Official Review · Reviewer_dn6h · 2024-05-15

**Rating:** 0
**Confidence:** 4

**Objective Part Of Review:**

Overview:
The paper proposes RUBRIC, a novel evaluation metric for information retrieval systems using LLMs and RAG. It addresses limitations of current methods by combining LLM efficiency with human judgment. RUBRIC decomposes queries into detailed questions, assesses responses based on these questions, and leverages LLM for initial grading, while humans refine and verify. Experiments on TREC benchmarks demonstrate its effectiveness and scalability.

Key Claims and Observations
• RUBRIC uniquely integrates LLMs and human judgment, efficiently processing data while ensuring evaluations reflect human notions of relevance and accuracy.
• Decomposing queries into answerable questions transforms relevance into tangible criteria, aligning with human cognition and allowing detailed assessment.
• RUBRIC is scalable and efficient, using LLMs to initially grade responses and supporting easy updates and extensions to the rubric.

Findings and Takeaways:
• Empirical validation on TREC benchmarks shows high correlation with traditional methods, confirming RUBRIC's potential as a reliable tool.
• The human-in-the-loop design emphasizes human expertise, enhancing reliability and mitigating risks of over-reliance on AI.
• RUBRIC can adapt to future changes in IR technologies and methodologies, incorporating new test collections and adjusting rubrics as needed.

• Problem Statement: The paper clearly states the problem of evaluating information retrieval systems using LLMs.
• Methods: The RUBRIC metric is comprehensively described, detailing the process of decomposing queries, grading responses, and calculating system scores.
• Results: Results are clearly stated and demonstrated using three TREC benchmarks, showing high correlation with traditional methods.
• Claims: Most Claims are supported with empirical evidence and comparisons to existing methods.
• Definitions/Abbreviations: To a large extent, concepts and notations are defined before use; however, the paper has room for improvement in this regard.
• Abstract/Introduction: Yes, they provide a clear overview of the goals and proposed solution, understandable without reading the full paper.
• Contradictions/Errors: There are some unclear or disconnected ideas in the paper across sections. These need to be put in a proper flow or removed so that the core idea of the paper is not muddled.
• Relevant Work Cited: There is extensive citation of relevant and competing works, showcasing a thorough review of the landscape.

## Immediate areas for improvement:

Some abbreviations and concepts are not defined on initial occurrence.
- Page 3: “1SL”, “DuoPrompt”, “SCU”
- Page 6: “AI2”, “ARC” and “TQA”

Some sentences are unclear as to their meaning or construction, or some citations are missing
- Page 3:
    - “In information retrieval evaluation there is no source text or gold summary to generate questions from. “
        - Is it “in the field of information retrieval evaluation”… “no standard or acceptable method” to generate questions?
    - “As part of the evaluation, our approach will operate in three phases (depicted in Figure 1), “
        - “our approach will operate” OR “our approach operates” (tense continuity)
- Page 7:
    - Which Table is being referenced by this line: “Best methods marked in bold-italics, methods within that range are considered equally good (marked in bold). “
    - Please cite reference [23] for this line “Since Sander’s work demonstrated “
    - Citation for FLAN-T5 missing
        - https://doi.org/10.48550/arxiv.2210.11416, {Scaling Instruction-Finetuned Language Models}
- Page 8:
    - We demonstrate that MRR- based Rubric Score is able to compare these methods on methods on the official leaderboard.
        - “methods on” or “methods with”?
- Page 9:
    - if possible please display Fig 3 (on page 5) nearer to its reference in the paper text (page 9)

## Immediate areas of _concern_

### HiTL descriptions are unclear and unconnected:
- [lends itself to HITL?] Abstract: “our para- digm lends itself for incorporating a human-in-the-loop without the danger of over-reliance on AI or resorting to expensive manual passage-level judgments. “
- [Human subject study in future work?] Human in the loop subsection in section “3 APPROACH”: “While our workbench software is designed to incorporating a human-in-the-loop,2 in this article we focus on the feasibility of the automatic part of this evaluation paradigm and demonstrating the verification process in Section 6.7. We leave the human subject study to future work. “
- Section 6.7 talks about Human in the loop verification: However, the section will benefit from an appropriate data analysis and a high level overview of the findings. The section currently reads as a list of cherry picked, albeit interesting observations, but the takeaways from this section are not clear in its current form.
- In any case, the HITL applicability of the RUBRIC method does seem plausible. However a deeper further study is warranted for sure.

### The relationship between FLAN-T5 and GPT4 is not clear.
- Maybe a figure or an algorithm to describe the relation between GPT and FLAN would help with understanding of the Method

### 6.5 Inter-Judge Agreement on TREC DL
- Cohen’s kappa agreement of 0.25 is generally considered as fair at best and not good, please provide more reasoning and justification for why 0.25 is good enough for this task.

### 6.4 Evaluating Text Generation Systems
- “We demonstrate that our RUBRIC approach can be used to evalu- ate systems that use natural language generation.”
    - I am not sure if this section is aligned with the core premise of this paper. However, the evaluation of NLG-based retrieval systems using this method deserves further investigation for sure.

### Kendall's tau vs Spearman’s rho
- “(1) our pro- posed approach can reproduce official TREC leaderboards nearly perfectly (Spearman’s rank of 0.97); “
    - If the goal of evaluation is to ensure that the true best systems are ranked appropriately, using Kendall’s tau to check for rank order is recommended as it captures the ordinal and monotonic comparisons best. Do your results change significantly if you use Kendall’s tau instead of Spearman’s rho?
    - Both Kendall's Tau and Spearman's Rank Correlation are used to measure the association between two ordinal variables. However, there are subtle differences to consider when choosing between them:
    Kendall's Tau (τ):
    •	Interpretation: Measures the strength and direction of the monotonic association between two variables. It focuses on the proportion of concordant and discordant pairs in the data.
    •	Statistical Properties: Has better statistical properties than Spearman's rho, especially with smaller sample sizes. Its p-values are considered more accurate.
    •	Robustness: Less sensitive to outliers and errors in the data.

## Suggested areas for improvement

The paper does hinge on an underlying assumption, albeit reasonable, that “We believe that the task of breaking a larger information need into the set of concise ques- tions is more natural for humans to accomplish than to directly judge the relevance of text. The process is akin to designing a grad- ing rubric for essay grading. Educators routinely break down com- plex assignments into specific criteria or questions, allowing for a more objective and detailed assessment of student work. “
- This assumption might benefit from a citation from the cognitive sciences field.

A related area to check out research on the informational, navigational, and transactional information needs, and if the author’s core assumption bears out for the different types of information needs and information seeking behaviors.
1. **Information Retrieval on the Web** , K. Yang (2005): Discusses informational, navigational, and transactional types of information needs in web search.
2. **Interactive Information Retrieval**  I. Ruthven (2008):   Covers the concepts of informational and transactional queries, as well as service finding in web search.
3. **Looking for Information: A Survey of Research on Information Seeking, Needs, and Behavior** , D.O. Case, L.M. Given (2016): Explores active information seeking, search behaviors, and different types of information needs.
4. **Is Exploratory Search Different? A Comparison of Information Search Behavior for Exploratory and Lookup Tasks** K. Athukorala, D. Głowacka, G. Jacucci, et al. (2016)  Analyzes different information needs including exploratory and transactional queries.

Was any experimentation done for comparing LLM performances?: It would be interesting to see how some of the open source LLMs perform for this same task

For the prompts listed in table 1:
-  what other prompts were experimented with? how were the returned RUBRIC questions evaluated? How did you know that the current prompts are reasonable enough to run further experiments?

**Subjective Part Of Review:**

•  Readability and Understandability: The paper is well-written to a large extent, with clear problem statement, methods, and results. However, the understandability can be improved with algorithms or diagrams describing the method.

•  Relevance of the problem: The problem addressed by the paper is highly relevant. As information retrieval systems, particularly those augmented by large language models, become increasingly complex, traditional evaluation methods struggle to keep pace. The paper addresses a critical need in the IR community for more scalable, efficient, and effective evaluation methods that reduce reliance on extensive manual labor while still incorporating human judgment.

•  Originality: The RUBRIC metric introduces an original approach to evaluating information retrieval systems by combining automated processes with human oversight. The method of decomposing queries into a set of detailed questions and using these to assess the relevance and completeness of system responses is innovative. This approach not only leverages the capabilities of large language models but also maintains human involvement in the evaluation process, striking a balance between automation and human insight.

•  Interest of the Results: The results are interesting and demonstrate the potential of the RUBRIC metric to effectively evaluate information retrieval systems. The paper shows that the RUBRIC approach correlates well with traditional evaluation paradigms using TREC benchmarks, suggesting that it can serve as a viable alternative to more labor-intensive manual methods. The ability to extend and reuse the test collections also adds significant value to the approach.

•  Interest to ICTIR Community: The paper is likely to be of high interest to the ICTIR community. The challenge of efficiently evaluating complex IR systems is a pressing issue, and the community is actively seeking solutions that can handle the nuances introduced by advanced AI technologies. The integration of LLMs with human judgment in a structured evaluation framework addresses a topical area of research and practice in information retrieval. The method's scalability, repeatability, and potential to reduce the burden of manual evaluation while maintaining quality are particularly compelling aspects that would attract attention from researchers and practitioners alike.

• Impact: RUBRIC offers a scalable, repeatable, and less subjective evaluation method for complex IR systems. Its integration of human insights and LLMs addresses a crucial need in the era of advanced AI, potentially setting a new standard for evaluation and influencing future research.

---

### Official Review · Reviewer_1dX7 · 2024-05-16

**Rating:** 1
**Confidence:** 5

**Objective Part Of Review:**

The problem is clearly stated: the authors propose a method for automatic evaluation of both ranked retrieval results and generated outputs using LLMs comparable to TREC manual assessment.

The methods are clearly described.  It seems clear which LLMs are used where, and what the prompts were.  This would benefit from a github repo or some kind of appendix, since presumably others would want to reuse the prompts and models.

The introduction makes several claims that are not supported in the paper: the evaluation is claimed to be "unbiased", but no bias is explored or measured.  It is also claimed to be "efficient" but efficiency is not treated in any way, aside from a comment that computing part of the results only took an hour.  The authors believe that writing a grading rubric is easier than making relevance assessments; perhaps this is not a claim, but at any rate it isn't addressed, since the authors did not conduct any relevance assessments in this work.  The method is claimed to be "dynamic", which is not even defined, and "scalable", which again ia not measured.

I found the concepts and notations are reasonably well defined.  The abstract and conclusion were clear to this reviewer.  When characterizing TREC relevance in the introduction, the authors are contradicted by the early chapters of the 2005 TREC book.  Relevance in TREC is never "abstract"... abstractness was a prime argument against using relevance as a criterion in the days before TREC (see Saracevic's very long paper on the topic), and for this reason and others TREC relevance has a very concrete definition.

The results state that the proposed method "reliably obtains the best results" (caption to table 3), but this is only true for Spearman correlations; according to Kendall's tau there are a number of cases where other methods have the best results.

The authors write "Significance tests do not apply to these rank correlation measures" but in fact there are tests associated with these correlations, although perhaps the tool they used to compute the correlations doesn't produce p-values.  At any rate, correlations are significant when they are above 0.3 or so, depending on the number of items correlated, so significance of a correlation is not a good indicator of "equivalent for evaluation".  (as indeed the definition of statistical significance says)

I was confused when the paper described how documents were cut into passages, because the TREC DL collection is already segmented into passages.  Did they resegment the documents?  Did they realize the judgments were made on the passages and then extrapolated to the documents much as they do here?

The related work is identified.  (warning, subjective part) I think the treatment here deserved more space, because a lot of the methods in all these papers, the present one included, could be mixed and matched to produce other methods like these which might themselves perform better on other models.

Section 6.7 on manual verification of LLM grades is very telling.  According to the paper it is frequently the case that the model in fact outputs that are not answers to the rubric questions.  This would imply we need to tune the questions to a model much as we might "engineer" a prompt, or perhaps that we need to use a minimum number of questions to avoid these cases dominating the final voted score.

Since the manual verification section addresses a specific set of retrieved documents for one DL query, I assume that these spurious outputs and their accompanying grades are included in the max-vote described in equation (1).  The verification section does not say what grades were given with these spurious answers.  Of course given that this is LLM output there is no actual causal link between the produced answer and the produced grade, so perhaps this is not important.

**Subjective Part Of Review:**

The authors posit that RUBRIC collections are more reusable because the grading rubrics can be updated when new relevant documents are found.  This would clearly bias evaluation results towards the system producing those documents.

In figures 2 and 3, "generated grading rubrics" are shown, but the text seems to imply that after the rubrics are generated, they are reviewed and refined manually by a person.  Are the rubrics shown the generated ones, or the edited ones?

Section 4.2 states that certain elements of the prompt text are "critical", but does not explain why or what this means.  Since (as they write) the prompt was suggested by an LLM, how do they know it is critical?

Footnote 5 says that they were unable to install the model for EXAM, but EXAM baselines were still produced.  I'm not sure how that can be... if they couldn't use the model, how were the results produced?  Perhaps they used another model that they could install?

A comparison is made to Thomas et al's prompts, which are significantly longer and were developed and refined on their own version of GPT-4.  Might that argue that the prompts are not ideal for the models used by the authors of this paper?

I would think that just as we worry about assessor agreement, we would also worry about model agreement, not just agreement of the model to the assessor, but agreement among models.  Why were these particular models (GPT-3.5, FLAN-T5) chosen for this paper?

---

### Official Review · Reviewer_AHR8 · 2024-05-23

**Rating:** -1
**Confidence:** 4

**Objective Part Of Review:**

In this paper, the authors propose an alternative evaluation approach that integrates human judges. Some experiments have been conducted on three TREC data sets and experimental results demonstrate that (1) their proposed approach can reproduce official TREC leaderboards; and (2) it is a strong contender in comparison to other recent LLM-based relevance label predictors.

**Subjective Part Of Review:**

To my understanding, the core idea of this paper is to convert a query into multiple subqueries and then enhance the results based on these subanswers. This idea is a normal routine for RAG which is not new. After reading this paper carefully, I am still not very clear what the unique contribution of this paper is. Lack of novelty is my concern for this paper. In addition, this paper is more like an experimental report than a research paper for proposing a new method.

The writing of this paper is good. However, some typos can be found in the paper. For example, on page 5, "We consider each each question 𝑟 on the rubric" ==> "We consider each question 𝑟 on the rubric".

---

### Meta-Review · Area_Chair_vygv · 2024-05-30

**Recommendation:** Accept (Oral)
**Confidence:** 4

**Metareview:**

The authors propose an LLM-based method for the automatic evaluation of search results in form of generated texts but also for ranked results. The idea is related to the previous EXAM metric in that it basically also asks how many questions related to a given information need can be answered from a search result (generated text or ranked documents).

Automatic evaluation attracted quite some interest recently, so that the paper is timely. The presented idea in general also is interesting. For my metareview, I disregard the 'weak reject' review as it somewhat indicates that the reviewer misinterpreted the paper's contribution ("the core idea of this paper is to [...] enhance the results based on these subanswers [...] normal routine for RAG").

Overall, I suggest to accept the submission but would hope that some important points will be clarified / included in a final version (cf. the individual comments in the reviews). Most importantly, the current experiments focus solely on the correlation of evaluation results using the newly proposed scheme to evaluation results based on previous schemes. Given that there is some level of correlation, the next natural question is how much effort is needed to run the evaluation with RUBRIC compared to other ideas. Not having anything on that end at all in the paper really is a pity. The paper could easily be a clear 'accept' candidate with effort-oriented results. In case of "competetiveness" to for instance traditional TREC assessor effort the applicability would be indicated, in case of more effort than for traditional schemes, the RUBRIC idea could be an interesting starting point to improve the required human effort in future work. I really hope that at least some results on the effort / efficiency angle will be included in case of acceptance as potential readers of the paper will really want to learn about the required effort.